# A Comparative Study on the Effects of Different Parts of *Panax ginseng* on the Immune Activity of Cyclophosphamide-Induced Immunosuppressed Mice

**DOI:** 10.3390/molecules24061096

**Published:** 2019-03-20

**Authors:** Li-xue Chen, Yu-li Qi, Zeng Qi, Kun Gao, Rui-ze Gong, Zi-jun Shao, Song-xin Liu, Shan-shan Li, Yin-shi Sun

**Affiliations:** 1Institute of Special Animals and Plants Sciences, Chinese Academy of Agricultural Sciences, Changchun 130112, China; 82101172456@caas.cn (L.-x.C.); 15568871905@163.com (Y.-l.Q.); GaoK2018@163.com (K.G.); grz2017caas@163.com (R.-z.G.); shaozijun2017@163.com (Z.-j.S.); liusx2018@163.com (S.-x.L.); Lishanshan@caas.cn (S.-s.L.); 2College of Chinese Medicinal Materials, Jilin Agricultural University, Changchun 130118, China; 3School of Pharmaceutical Sciences, Jilin University, Changchun 130021, China; qizeng95@163.com

**Keywords:** cyclophosphamide, immunosuppression, different parts of ginseng, immunomodulatory effect

## Abstract

The objective of the present study was to compare the effects of the immunological activity of various parts (root/stem/leaf/flower/seed) of five-year-old ginseng on the immune system of immunosuppressive mice. Immunosuppression was induced by cyclophosphamide (CTX) in the mouse model, whereas levamisole hydrochloride tablet (LTH) was used for the positive control group. We found that ginseng root (GRT), ginseng leaf (GLF), and ginseng flower (GFR) could relieve immunosuppression by increased viability of NK cells, enhanced immune organ index, improved cell-mediated immune response, increased content of CD4^+^ and ratio of CD4^+^/CD8^+^, and recovery of macrophage function, including carbon clearance, phagocytic rate, and phagocytic index, in immunodeficient mice. However, ginseng stem (GSM) and ginseng seed (GSD) could only enhance the thymus indices, carbon clearance, splenocyte proliferation, NK cell activities, and the level of IL-4 in immunosuppressed mice. In CTX-injected mice, GRT and GFR remarkably increased the protein expression of Nrf2, HO-1, NQO1, SOD1, SOD2, and CAT in the spleen. As expected, oral administration of GRT and GFR markedly enhanced the production of cytokines, such as IL-1β, IL-4, IL-6, IFN-*γ*, and TNF-*α*, compared with the CTX-induced immunosuppressed mice, and GRT and GFR did this relatively better than GSM, GLF, and GSD. This study provides a theoretical basis for further study on different parts of ginseng.

## 1. Introduction

*Panax ginseng* C. A. Meyer (*P. ginseng*), known as a traditional herbal medicine, has been used for thousands of years to treat medical illnesses or to maintain homeostasis of the body [1,2,3]. It is also one of the most popular medicinal herbs with tonic effects in China and other countries [4]. Ginseng contains various compounds, such as saponins, polysaccharides, and polypeptides. Many modern studies indicate that ginsenosides are the main secondary metabolites of *P. ginseng*, and they are thought to be the major bioactive components responsible for ginseng’s restorative functions [5,6,7]. Numerous studies have been conducted to test the effects of *P. ginseng* on systemic immune function [8,9]; its anticancer [10], antidiabetic [11], antiaging [12], renal protective [13], and antidepressive [14] effects; and its effects on sexual [15] and brain function [16]. Several bioactive ingredients, including polysaccharides, fatty acids, peptidoglycans, and various individual ginsenosides, have been identified in *P. ginseng* [17,18]. In the past few years, considerable research has been undertaken to measure and analyze the ingredients and effects of ginseng. The root of ginseng is considered to be the most active part of the ginseng plant, but there has been no significant research on other plant parts, such as the stem, leaves, flowers, and seed [19].

In addition to the root, the stems, leaves, flowers, and seeds of *P. ginseng* have also been found to be rich in saponins. These parts have a similar chemical composition and pharmacological activities as the root [20,21]. The pharmacologic function of ginseng stem-and-leaf saponins is extensive and applies to multiple systems of the body, mainly in regulating the body’s immune function, regulating antitumor and antiaging response, promoting cell proliferation, and regulating the endocrine and respiratory systems [22,23]. In major ginseng cultivation countries, many ginseng stems and leaves are discarded as waste in the field and are not fully utilized [24]. Intensive study of the pharmacological effects of ginseng stem-and-leaf saponins and exploration of the mechanism of action will provide a reference for further development and utilization of ginseng stem-and-leaf resources, which have important application value and have broad prospects. The total saponin content of ginseng flowers is higher than that of the roots. In particular, the levels of ginsenosides Rd and Re have been found to be greater in the flowers compared with the roots, although the levels of other ginsenosides are similar [25]. Numerous studies have demonstrated that ginseng flowers can enhance human immune function, increase the synthesis of stem cell proteins and nucleotide adenine dinucleotide (NAD), and significantly inhibit the growth of cervical cancer cells [19]. Therefore, the medicinal and health value of ginseng flower buds may not be lower than that of ginseng roots. Ginseng seeds are rich in unsaturated fatty acids, mainly oleic acid and linoleic acid [26]. Despite extensive research reporting that unsaturated fatty acids and phytosterols have therapeutic effects on metabolic syndrome by lowering cholesterol and improving insulin sensitivity [27], the pharmacological activity of ginseng seeds has not been fully explained; in particular, the immunocompetence of ginseng seed has rarely been reported. Therefore, in order to explore the potential clinical value of different parts of ginseng, we compared the effects of extracts from various parts of five-year-old ginseng on the immunological activity of an immunosuppressed mouse model.

## 2. Results

### 2.1. Effect of Different Parts of Ginseng on Body Weight and Immune Organ Indices of Mice

The model control (cyclophosphamide, CTX), levamisole hydrochloride tablet (LTH), ginseng root (GRT), ginseng stem (GSM), ginseng leaf (GLF), ginseng flower (GFR), and ginseng seed (GSD) groups were subjected to intraperitoneal injection of cyclophosphamide at a dosage of 50 mg/kg on days 26–29, and mice were orally administered different parts of ginseng for 30 days. Spleens were collected immediately from the sacrificed mice under aseptic conditions. Oral administration of the different parts of ginseng for 30 days did not result in any mortality or in any significant reduction in final body weight of mice, indicating that the ginseng formulations did not induce toxic effects in mice. The thymus and spleen are vital immune organs and are accountable for initiating immune reactions in the body. The CTX group (CTX 50 mg/kg) showed a significant decrease in body weight and in spleen and thymus indices compared with the control group (*p* < 0.05). The group treated with LTH and the GSD group had significantly higher body weights than mice in the CTX group (*p* < 0.05). The spleen index values of the GRT and GFR groups were significantly higher than that of the CTX group (*p* < 0.01). Compared with the CTX group, there were observable increases in the thymus index values in the GRT, GLF, GFR, and GSD groups (*p* < 0.05) (Table 1). 

### 2.2. Effect of Different Parts of Ginseng on Cellular Immunity

#### 2.2.1. Effect of Different Parts of Ginseng on the Concanavalin A (ConA)-Induced Splenocyte Proliferation 

Compared with the control group, the CTX group had significantly decreased concanavalin A (ConA)-induced splenocyte proliferation (Figure 1A) (*p* < 0.01). The LTH, GRT, and GFR groups showed significantly higher ConA-induced proliferation of splenic lymphocytes than the CTX group (*p* < 0.05, *p* < 0.01), whereas the GSM, GLF, GFR, and GSD groups showed no remarkable difference compared with the GRT group (*p* > 0.05). 

#### 2.2.2. Effect of Different Parts of Ginseng on Delayed-Type Hypersensitivity (DTH) to Sheep Red Blood Cells (SRBC)

As can be seen from Figure 1B, the CTX group had lower footpad thickness than the control group (*p* < 0.05). The LTH group and the GST-, GLF-, and GFR-treated groups showed a significantly higher delayed-type hypersensitivity (DTH) degree than the CTX group (*p* < 0.05). The groups of mice treated with GSM and GSD showed no significant difference in footpad thickness (*p* > 0.05). The DTH degrees in the GSM, GLF, and GSD groups were markedly lower than in the GRT group, while no change was observed in the GFR group (*p* > 0.05).

### 2.3. Effect of Different Parts of Ginseng on Leukocyte Count

The CTX group had a significantly lower leukocyte count than the control group (Figure 1C) (*p* < 0.05). The leukocyte counts of the GRT, GLF, and GFR groups were significantly larger (*p* < 0.05) than that of the CTX group and relatively higher than those of the GSM and GSD groups (*p* > 0.05). The GSM, GLF, and GSD groups had markedly downregulated leukocyte counts compared with the GRT group, and the leukocyte count of the GFR group was slightly, but not significantly, higher than that of the GRT group (*p* > 0.05).

### 2.4. Effect of Different Parts of Ginseng on Macrophage Phagocytosis

#### 2.4.1. Effect of Different Parts of Ginseng on Carbon Granular Clearance Assay

According to the results of carbon granular clearance assay (Figure 2A), macrophage phagocytosis was significantly reduced by 22% in the immunosuppressed mice compared with the control group mice (*p* < 0.05). The LTH, GRT, GSM, GLF, GFR, and GSD groups showed the highest index values of macrophage phagocytosis, which were significantly increased by 25%, 24%, 18%, 18%, 25%, and 24%, respectively, in comparison with the CTX group (*p* < 0.05, *p* < 0.01), while the GSM, GLF, GFR, and GSD groups showed no observable difference compared with the GRT group (*p* > 0.05).

#### 2.4.2. Effect of Different Parts of Ginseng on the Phagocytic Capacity of Macrophage Cells

According to the phagocytic capacity assay conducted on macrophage cells, the phagocytic rate in the CTX group had remarkably decreased by 14% in comparison to the control group (Figure 2B) (*p* < 0.01). Compared with the CTX group, the LTH, GRT, GLF, and GFR groups showed significant increases in phagocytic rate by 12.6%, 17.8%, 9.15%, and 18.2%, respectively (*p* < 0.05, *p* < 0.01). The GSM and GSD groups showed no significant difference in phagocytic rate compared with the CTX group (*p* > 0.05). The phagocytic index of the CTX group was significantly lower by 24% than that of the control (Figure 2C) (*p* < 0.01). The phagocytic index values of the LTH, GRT, GFR, GLF, and GSD groups (but not the GSM group) were significantly higher than that of the CTX group by 14%, 21%, 22%, 19%, and 12%, respectively (*p* < 0.05, *p* < 0.01). 

### 2.5. Effects of Different Parts of Ginseng on Splenic NK Cell Activity

NK cells play a key role in killing tumor cells. Compared with the control group, the CTX group showed a significant decrease in NK cell activity of 37% (*p* < 0.05), as shown in Figure 2D. The GRT, GSM, GLF, GFR, and GSD groups showed notably enhanced (41%, 21%, 27%, 35%, and 23%, respectively) NK cell activity compared with the CTX group (*p* < 0.05, *p* < 0.01). Comparison between the GSM, GLF, GFR, GSD, and GRT groups showed no marked difference in NK cell activity (*p* > 0.05).

### 2.6. Effect of Different Parts of Ginseng on Splenic T-Lymphocyte Subpopulations

As ginseng treatment increased the activity of T cells by altering the quantities of T cells or their subpopulations, we conducted a phenotypic analysis of the total T cells and the T cell subsets (Figure 3). The content of CD4^+^CD8^−^ in the CTX group was significantly downregulated, by 14.89%, compared with the control group (*p* < 0.01). The LTH, GRT, GLF, and GFR groups showed significantly enhanced CD4^+^CD8^−^ content when compared with the CTX group (*p* < 0.05, *p* < 0.01). The content of CD4^+^CD8^−^ in the GSM and GSD groups was not significantly different from that of the CTX group (*p* > 0.05). Compared with the GRT group, the levels of CD4^−^CD8^+^ were not obviously different in the GSM, GLF, GFR, and GSD groups of mice. The ratios of CD4^+^/CD8^+^ in the control, LTH, GRT, GLF, GFR, and GSD groups were dramatically enhanced compared with the CTX group (*p* < 0.05, *p* < 0.01). The GSM and GSD groups showed a remarkable decrease in the CD4^+^/CD8^+^ ratio compared with the GRT group. The CD4^+^CD25^+^ content in the control, LTH, GRT, GSM, GLF, and GFR groups were lower than that in the CTX group (*p* < 0.05, *p* < 0.01).

### 2.7. Effects of Different Parts of Ginseng on Cytokine Concentrations in Serum

In this study, the CTX group showed markedly decreased levels of IL-1β, IL-4, IL-6, IFN-γ, and TNF-α than the control (*p* < 0.05, *p* < 0.01). However, GRT and GFR treatment were able to recover the serum concentration of these cytokines in immunosuppressed mice (Figure 4). 

### 2.8. Effects of Different Parts of Ginseng on the Nrf2, HO-1, NQO1, SOD1, SOD2, CAT Signaling Pathway

In order to study the underlying mechanism of ginseng root and flower treatment in CTX-induced immunosuppressed mice, the expression levels of some antioxidative enzymes and some proteins involved in antioxidant pathways were measured. Mice treated with CTX exhibited markedly reduced expression levels of nuclear factor-erythroid 2 related factor 2 (Nrf2), heme-oxygenase-1 (HO-1), NADPH quinine oxidoreductase (NQO1), superoxide dismutase 1 and 2 (SOD1, SOD2), and catalase (CAT) in the spleen compared with the control (*p* < 0.01, Figure 5). Thirty days of LTH, GRT, and GFR administration markedly enhanced the Nrf2, HO-1, NQO-1, SOD1, SOD2, and CAT expression levels after immunosuppression in mice (*p* < 0.05, *p* < 0.01). Compared with the GRT group, the GFR group showed obviously upregulated expression levels of Nrf2, HO-1, and CAT (*p* < 0.05, *p* < 0.01), while the GFR group showed no remarkable regulatory effects on the expression levels of SOD1, SOD2, and NQO1 compared with the GRT group (*p* > 0.05).

## 3. Discussion

Recently, ginseng roots, a traditional herbal medicine, have been considered for use as an immune regulator in numerous studies [28]. However, the immunological properties of various parts of the ginseng plant (i.e., stem, leaf, flower, and seed) are less studied. In this study, we compared the immune competence effects of various parts of five-year-old ginseng on immunosuppressed mice.

Cyclophosphamide, an inducer of immunosuppression, is one of the most commonly used malignant tumor agents in clinical chemotherapy, and its application often causes severe side effects, such as leukopenia, immunosuppression, and myelosuppression. Cyclophosphamide is an alkylating immunosuppressant that acts on the cell cycle by interfering with DNA and RNA functions, cross-linking DNA, and inhibiting DNA synthesis, which ultimately inhibits the proliferation of T and B lymphocytes [29,30]. Ginseng treatment can return immunocompromised mice back to normal immune levels by enhancing both specific and nonspecific immune responses and improving the efficacy of the immune system [31,32]. In addition, levamisole hydrochloride tablet, which is known to exert immune regulation and immune stimulatory effects [33,34], was used for the positive control.

In this study, we employed various assays to evaluate the effects of various parts of five-year-old ginseng on the innate and adaptive immune responses. The results of these assays indicate that the various parts of ginseng reduce the side effects of immune suppression in mice by enhancing cellular immunity, macrophage phagocytosis, leukocyte count, and NK cell activity. However, different ginseng parts exhibited different activities in immunocompromised mice. The mechanism underlying these activities may be related to the activation of Th cells, cytokine responses, and protein expression. 

T cells and B cells, which primarily mediate cellular immune response and humoral immune response, are most effective in resisting bacteria, viruses, and cancer cells that present tumor antigens [35,36]. A study demonstrated that IL-6 plays a considerable role in improving T cell activation and B cell differentiation [37]. T-helper type 1 cells suppress the pleiotropic effects of IFN-γ, such as the regulation of the immune response and the regulation and growth of macrophages, NK cells, and T and B cells [38,39]. Studies have suggested that IL-4 and Batf form a positive feedback amplification loop to induce Th2 cell differentiation and the subsequent Th2-type immune response. Bach2–Batf interactions are required to prevent excessive Th2 response [40], while IL-4 regulates the secretion of lymphocytes and macrophages. The immunostimulatory effect of ginseng may be useful for treating diseases in which the pathology is dependent on reduced phagocytotic capacity, such as Crohn’s disease [41,42]. In CTX-injected mice, GRT, GLF, and GFR remarkably increased the viability of NK cells, enhanced the immune organ index, and improved the cell-mediated immune response, including splenocyte proliferation and delayed-type hypersensitivity. At the same time, the GRT, GLF, and GFR groups recovered macrophage function, including carbon clearance, phagocytic rate, and phagocytic index, in immunosuppressed mice. On the other hand, the GSM and GSD groups moderately relieved immunosuppressed mice in thymus indices, carbon clearance, splenocyte proliferation, NK cell activities, and the level of IL-4. There was a significant enhancement of the macrophage phagocytic function in the GLF and GSD groups after CTX challenge, which may be due to higher levels of IL-4. GRT and GFR recovered the profound depression induced by CTX via upregulation of the levels of IL, IFN, and TNF and did so relatively better than the GSM and GSD. The immune effects of the GRT, GLF, and GFR groups were better than those of the GSM and GSD groups, which may be due to the significantly higher ginsenosides (Appendix A. The fingerprint of eight individual ginsenoside in five-year-old ginseng of various parts; Appendix A. The content of ginsenosides in five-year-old ginseng of various parts.) and saponins and mino acid content (Appendix A. The content of amino acids in five-year-old ginseng of various parts; Appendix A The content of total saponins and total polysaccharide in various parts ginseng) of the GRT, GLF, and GFR groups compared with the GSM and GSD groups. In particular, the content of individual ginsenosides Rg_1_, Re, Rb_1_, Rc, and Rb_2_ in the GRT, GFR, and GSD groups were remarkably higher than in the GSM and GSD groups. A previous research also demonstrated that Rg1, Re, and Rb1 have more potent adjuvant properties, indicating that they are major contributors to the adjuvant activities of total ginseng saponins [43]. 

Appropriate ratio and counts of CD4^+^ and CD8^+^ T-lymphocyte subpopulations are key to immunoregulation. A disequilibrium of T-lymphocyte subgroups can lead to immune dysfunction, resulting in a series of immunopathological changes that affect the body’s immune protection mechanisms [35]. Based on the different cytokines they produce, CD4^+^ T cells can be further differentiated into Th1 and Th2 cells. The former is mainly responsible for secreting IL-2, IL-12, IFN-γ, and TNF, while the latter is mainly responsible for secreting IL-4, IL-5, IL-6, and IL-10 [44]. In this study, treatment with CTX decreased CD4^+^ and the CD4^+^/CD8^+^ ratio as well as the levels of serum cytokines, while GRT, GLF, and GFR improved the CD4^+^ cell count and the CD4^+^/CD8^+^ ratio, indicating that immune function was recovered by oral administration of GRT, GLF, or GFR.

Numerous studies have demonstrated that antioxidant enzymes, including SOD, CAT, and GSH-Px, play a considerable role in protecting against free radical damage and oxidative stress, thereby relieving immunosuppression [45]. The transcription factor Nrf-2, a key transcription factor in cellular antioxidant stress systems, induces the transcription of numerous antioxidant and cellular defense genes, including NADP(H), NQO-1, HO-1, SOD, CAT, and glutathione peroxidase (GPx) [46]. Nrf2/HO-1 and Nrf2/NQO-1 have an important positive effect on oxidative damage and protection of the body. When the body is in an oxidative stress state, Nrf2 induces high expression of HO-1 and NQO-1, protecting the body from damage. The effects of GRT and GFR treatment on immunosuppressed mice were better than those of GSM, GLF, and GSD. Therefore, we studied the effects of GRT and GFR on the expression of Nrf2, HO-1, NQO1, SOD1, SOD2, and CAT proteins. In this study, CTX inhibited the expression of Nrf2, SOD1, SOD2, HO-1, CAT, and NQO1, which may have resulted in the profound depression of immune-supporting activity. Our study suggests that Nrf2/HO-1 and Nrf2/NQO1 signaling play a crucial role in GRT- and GFR-mediated protection and antioxidation against CTX-induced immunosuppression.

## 4. Materials and Methods

### 4.1. Reagents

RPMI-1640 medium, fetal bovine serum (FBS), benzylpenicillin, and streptomycin were offered by Gibco (Grand Island, NY, USA). Dimethyl sulfoxide, ConA, trypan blue, and 5-diphenyl-tetrazolium bromide and 3-(4,5-dimethylthiazol-2-yl)-2 (MTT) were provided by Sigma-Aldrich Co., (St Louis, MO, USA). YAC-1 cells were obtained from the Cell Bank of the Chinese Academy of Sciences (Shanghai, China). Cyclophosphamide was provided by Shengdi Pharmaceutical Co., Ltd. (Jiangsu, China). Levamisole hydrochloride tablets were obtained from Renhetang Pharmaceutical Co., Ltd. (Shandong, China). The antibodies for the T cell subpopulations assay were provided by BioLegend (San Diego, CA, USA), including fluorescein isothiocyanate (FITC)-conjugated anti-mouse CD4, allophycocyanin (APC)-conjugated anti-mouse CD8a, and phycoerythrin (PE)-conjugated anti-mouse CD25. Cytokines, including IL-1β, IL-4, IL-6, IFN–γ, and TNF-α, were provided by Invitrogen Co., Ltd. (Carlsbad, CA, USA). In this study, anti-Nrf2, anti-HO-1, anti-NQO1, anti-SOD1, anti-SOD2, and anti-CAT antibodies were provided by Abcam (Cambridge, MA, USA).

### 4.2. Preparation of Ginseng Extracts of Different Parts

The different parts of five-year-old ginseng plant (root/steam/leaf/flower/seed) were obtained from Zhongsen Pharmaceutical Co., Ltd. (Jilin, China) and identified by Professor Wei Li of the College of Chinese Medicinal Materials, Jilin Agricultural University. The ginseng was extracted according to a previously detailed method [47]. First, 100 g of crushed ginseng material was placed in a beaker and then sonicated at room temperature (25 ± 1 °C) for 30 min. The ginseng extraction was performed three times under reflux with distilled water; the extract was concentrated to 200 mL and freeze-dried into a powder. The extract powders were suspended in deionized water in concentrations of 10 mg/mL and stored in a refrigerator at 4 °C. 

### 4.3. Experimental Animal and Treatment

In this experiment, BALB/c mice (male, 18–22 g) provided by Changsheng Biotechnology Co., Ltd. (Liaoning, China), were selected as research objects. These mice were separately caged (5 mice per cage) and raised in a standard lab environment (12 h light/dark cycle, 23 ± 1 °C, relative humidity 50 ± 5 %) with free access to food and water and then fasted 12 h prior to the experiment with free access to water only. All experiments were conducted in strict accordance with the guidelines of Care and Use of Laboratory Animals. 

After one week of acclimatization, animals were randomly divided into eight groups (*n* = 10): (1) control, (2) CTX (50 mg/kg), (3) LTH, (4) GRT, (5) GSM, (6) GLF, (7) GFR, and (8) GSD. The control group was given deionized water for 30 days. The model control (CTX), LTH, GRT, GSM, GLF, GFR, and GSD groups were subjected to intraperitoneal injection of cyclophosphamide at a dosage of 50 mg/kg on days 26–29. The positive control (LTH) group consisting of an immunosuppressed mouse model was intragastrically administered LTH (40 mg/kg) for one month. The mice of the remaining five groups were intragastrically administered the corresponding various ginseng extracts (GRT/GSM/GLF/GFR/GSD), and the dosage of administration was 1.0 g/kg. After 30 days of medication, the mice were euthanized by the method of cervical dislocation. A blood sample from the ophthalmic venous plexus was collected. Then, the spleens and thymuses were rapidly isolated and weighed, and the indices (organ weight/body weight ratio) were calculated.

### 4.4. Statistical Analysis

Significant intergroup differences were analyzed using one-way analysis of variance, and a post-hoc test was performed for intergroup comparisons using Bonferroni multiple comparison with the GraphPad Prism 6.0 software (CA, USA). Data are expressed in the form of mean ± standard deviation (SD).

### 4.5. Experimental Assays

#### 4.5.1. SRBC-Induced DTH

Mice (10 per group) were subjected to intraperitoneal injection of 50 mg/kg cyclophosphamide from day 26 for 4 days to induce immunosuppression. One hour after injection, the mice were intraperitoneally injected with 0.2 mL 2% defibrinated SRBCs (1 × 10^8^ cells) for the DTH assay. On day 30, a vernier caliper was used to measure the baseline footpad thickness of the left rear foot of 150 mice. Subsequently, each mouse was given a subcutaneous injection of 20 μL of 20% (v/v) SRBCs (1 × 10^8^ cell) into the left rear footpad, and the thickness of the rear footpad was remeasured after 24 h. The mean value of three measured results was used as the final result. The difference in footpad thickness indicated the effect of ginseng on cellular immunity.

#### 4.5.2. Carbon Granular Clearance Assay

Mice (10 per group) were injected intraperitoneally with 50 mg/kg cyclophosphamide from day 26 for 4 days to induce immunosuppression. On day 31, the mice were injected intravenously into the coccygeal vein with India ink 0.1 mL/10g that was diluted 4 times with sterile saline. Twenty-microliter blood samples were collected from the ophthalmic venous plexus at 2 and 10 min after the injection. Two milliliters of 0.1% sodium carbonate solution were mixed with the blood samples. The optical density (OD) values were measured at 600 nm by a multifunction microplate reader (xMark, USA). After blood sampling, the animals were euthanized, and the spleens and livers were isolated and weighed. The phagocytic index α was applied to validate the function of mouse phagocytes in clearing carbon particles, and it was calculated using the follow equation: *α* = body weight × $\sqrt[{3}]{k}$/(liver weight + spleen weight); k = (lg OD_1_ − lg OD_2_)/(T_2_ − T_1_).

#### 4.5.3. Phagocytic Function of Peritoneal Macrophage

On day 26, the mice (except for the normal control group) were administrated with cyclophosphamide (50 mg/kg, i.p.) daily for 4 days to induce immunosuppression. On day 31, the mice were immunized by intraperitoneal injection with 1 mL of 20% chicken red blood cells (CRBCs) and then sacrificed 30 min after the injection. Peritoneal cells were obtained from the peritoneal cavity using a peritoneal lavage, and the cells were then suspended in 2 mL of saline solution. A quantity of 1 mL of cell-rich lavage fluid was sucked and smeared onto glass slides before being incubated at 37 °C for 30 min. Nonadherent cells were washed off with saline solution, whereas the macrophage cells were fixed with an acetone–methanol mixture (1:1, v/v) before being stained with Giemsa–PBS solution (4%, v/v). After that, the stained cells were subjected to distilled water rinsing and air-drying before being counted with a 40× magnification microscope to calculate the phagocytic rate and the phagocytic index. The phagocytic rate was calculated as the percentage of macrophages that phagocytose CRBCs. The phagocytic index was calculated as the number of CRBCs that were phagocytosed per 100 macrophages.

#### 4.5.4. Splenocyte Proliferation Assay and Splenic NK Cell Activity Assay

Spleens were collected from the sacrificed mice under aseptic conditions. Then, erythrocyte debris and clumps were removed to obtain a single-cell suspension. Splenocytes were washed three times with PBS and then suspended in a solution of complete RPMI-1640 medium with a final density of 3 × 10^6^ cells/mL. Spleen cells were cultivated in 24-well plates without (control wells) or with 75 μL of ConA as a T cell stimulant and then incubated at 37 °C under humid 5% CO_2_ conditions for 68 h. After incubation, 0.7 mL of medium was discharged from each well, and then 0.7 mL of RPMI-1640 without FBS and 50 μL of 5 mg/mL MTT were added as a replacement. After incubation at 37 °C and 5% CO_2_ for 4 h, 1 mL of acidic isopropanol solution was added to each well to dissolve the insoluble purple formazan product. Subsequently, the OD values were measured at 570 nm using a microplate reader. Each measurement was performed three times. The proliferation capacity was characterized by the difference in absorbance between the splenocyte culture treated with ConA and that without ConA.

The splenic NK cell activity was evaluated by the lactic acid dehydrogenase (LDH) releasing method using YAC-1 cells as the target cells and setting the effecter/target cell ratio to 50:1. The spontaneous release level and the maximum release level of each well were examined. Quantities of 100 μL of target cells and 100 μL of Triton X-100 (2.5%, v/v) were added to the maximum releasing well. The test for each well was performed three times. After incubation for 4 h, the cells on each plate were subjected to centrifugation treatment at 1500 rpm for 5 min. Subsequently, 100 μL of supernatant was added to each well of a 96-well plate and incubated at 37 °C for 10 min. After that, 100 μL of substrate mixture was applied to each well. The absorbance was measured at 490 nm using a microplate reader. Using the following formula, the splenic NK cell activity (%) can be obtained:NK cell activity(%) = (OD_experimental_ − OD_spontaneous_) × 100/(OD_maximum_ − OD_spontaneous_)

#### 4.5.5. Mouse Leukocyte Assay and Splenic T-Lymphocyte Subpopulations Assay

For the mouse leukocyte assay, mice (10 per group) were injected intraperitoneally with 50 mg/kg cyclophosphamide from day 26 for 4 days to induce immunosuppression. On day 31, a 20 μL blood sample from the ophthalmic venous plexus was collected and diluted in 0.38 mL Turk’s solution, and the leucocytes were then counted by a microscope (URIT-2900Vet Plus).

For the splenic T-lymphocyte subpopulation assay, the splenocyte suspension was adjusted to 1 × 10^6^ cells/mL and subjected to flow cytometry to measure the splenocyte lymphocyte subpopulations. First, we labeled the splenocyte surface markers with PerCP-Cy5.5-conjugated anti-mouse CD3, FITC-conjugated anti-mouse CD4, APC-conjugated anti-mouse CD8a, and PE-conjugated anti-mouse CD25. We then washed the labeled cells twice, resuspended them in staining buffer (BioLegend), and finally analyzed them with FACSCalibur (BD, USA) and CellQuest software. As a comparison, we adopted the cells that were stained by isotype-matched antibodies to calibrate the settings of the FACSCalibur instrument.

#### 4.5.6. The Determination of Cytokines in Serum by ELISA

Blood samples were obtained from the ophthalmic venous plexus, then subjected to centrifugation treatment at 1006.2 ×g/RCF for 10 min before storing at −80 °C. The experiment was carried out using the ELISA double antibody sandwich method according to the ELISA Kit.

#### 4.5.7. Western Blotting Analysis

The spleen tissues were lysed using RIPA lysis buffer. The proteins were analyzed by 12% SDS-PAGE and transferred onto a polyvinylidene difluoride (PVDF) membrane. The membranes were washed with Tris-buffered saline (TBS) and blocked for 1.5 h with TBS containing 5% nonfat milk, then incubated with the primary antibodies against Nrf2 (1:1000), HO-1 (1:1000), NQO1 (1:800), SOD1 (1:500), SOD2 (1:500), and CAT (1:200) at 4 °C overnight. The membranes were then washed with TBS four times and incubated for 1 h at room temperature with secondary antibodies, washed with TBS and with Tween 20 (TBST) four times, and then developed. The intensity of the bands was assayed using Image J software (Media Cybernetics, Rockville, MD, USA).

## 5. Conclusions

In summary, GRT and GFR successfully improved organ indices, increased NK cell activity and macrophage function, enhanced cell-mediated immune responses and the ratio of CD4^+^/CD8^+^, and activated Th1 and Th2, which secrete cytokines in CTX-injected mice. The immunization of GRT and GFR was relatively better than those of GSM, GLF, and GSD. This study provides insight into the effects of GRT and GFR in CTX-induced immunodeficient mice related to the Nrf2/HO-1 and Nrf2/NQO1 signaling pathways. These results indicate that GRT and GFR have potential applications in the treatment of immunosuppressive diseases.

In addition, this study shows that, similar to the more commonly used ginseng root, the ginseng flower is beneficial for the recovery of the immune system of cyclophosphamide-induced immunosuppressed mice. In China, the utilization of the ginseng flower is minimal; thus, our research provides scientific evidence for the further development and potential clinical applications of the ginseng flower.

## Figures and Tables

**Figure 1 molecules-24-01096-f001:**
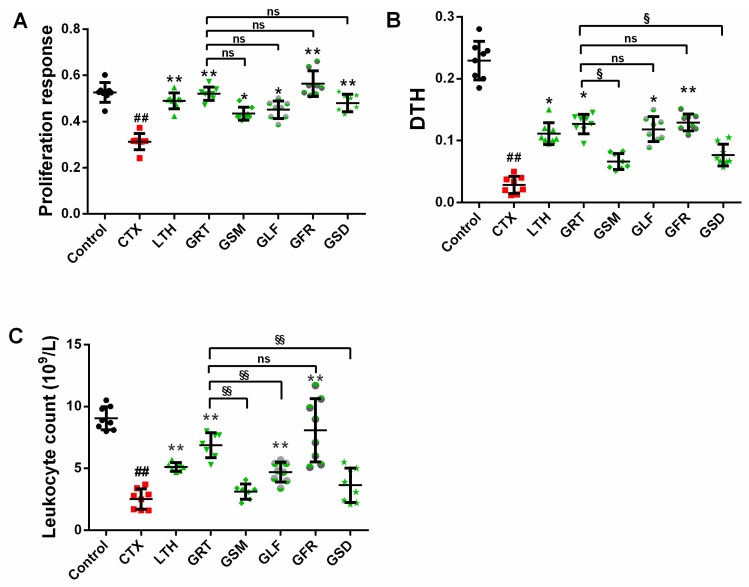
Effect of various parts of ginseng on the (**A**) concanavalin A (ConA)-induced splenocyte proliferation; (**B**) sheep red blood cells (SRBC)-induced delayed-type hypersensitivity (DTH); and (**C**) leukocyte count. The values are presented as mean ± SD, *n* = 10. ^##^
*p* < 0.01 compared with the control group. ** *p* < 0.01 and * *p* < 0.05 compared with the CTX group. ^§§^
*p* < 0.01, ^§^
*p* < 0.05, ^ns^
*p* > 0.05 compared with the GRT group.

**Figure 2 molecules-24-01096-f002:**
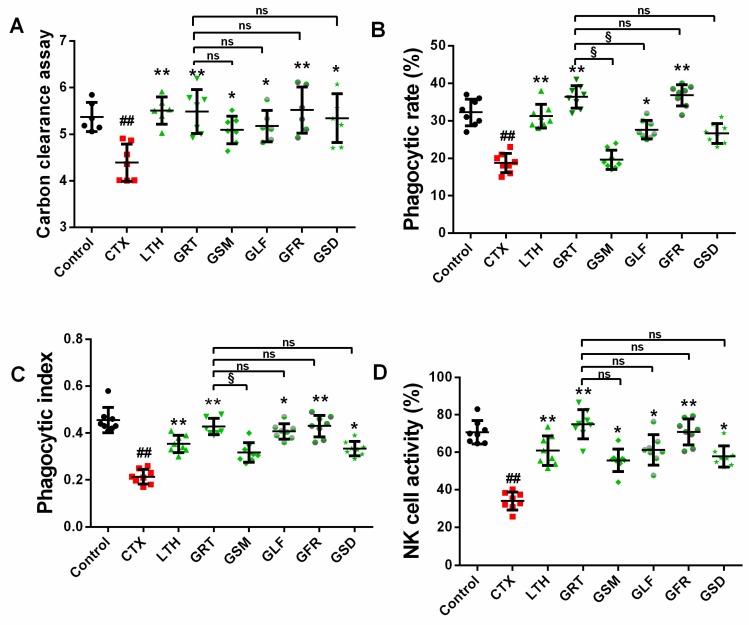
Effect of various parts of ginseng on the (**A**) carbon clearance assay; (**B**) phagocytic rate; (**C**) phagocytic index; and (**D**) NK cell activity. The values are presented as mean ± SD, *n* = 10. ^##^
*p* < 0.01 compared with the control group. ** *p* < 0.01 and * *p* < 0.05 compared with the CTX group. ^§^
*p* < 0.05, ^ns^
*p* > 0.05 compared with the GRT group.

**Figure 3 molecules-24-01096-f003:**
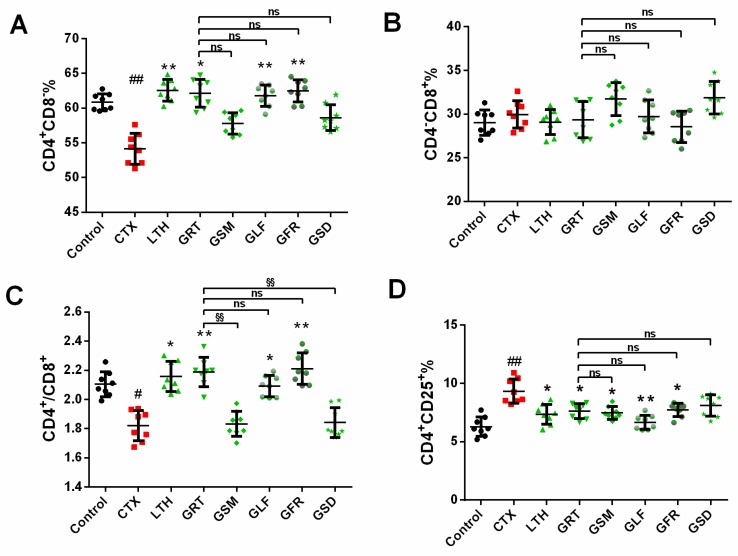
Effect of various parts of ginseng on the (**A**) CD4^+^CD8^−^%; (**B**) CD4^−^CD8^+^%; (**C**) CD4^+^/CD8^+^%; (**D**) CD4^+^CD25^+^%. The values are presented as mean ± SD, *n* = 10. ^##^
*p* < 0.01 and ^#^
*p* < 0.05 compared with the control group. ** *p* < 0.01 and * *p* < 0.05 compared with the CTX group. ^§§^
*p* < 0.01, ^ns^
*p* > 0.05 compared with the GRT group.

**Figure 4 molecules-24-01096-f004:**
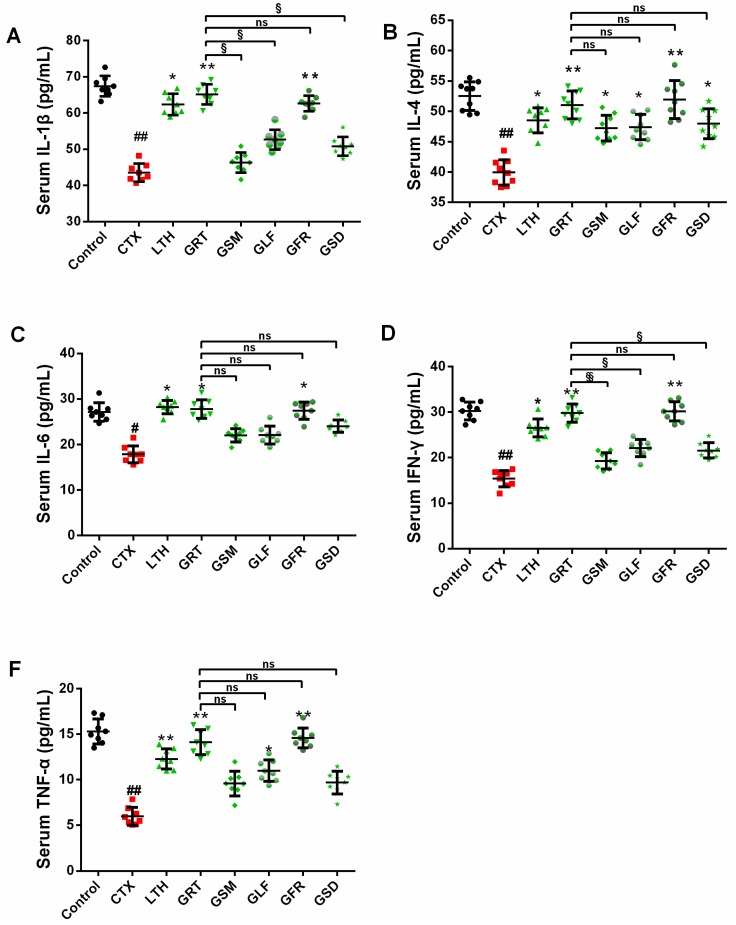
Effect of various parts of ginseng on the serum levels of (**A**) IL-1β; (**B**) IL-4; (**C**) IL-6; (**D**) IFN-γ; (**E**) TNF-α. The values are presented as mean ± SD, *n* = 10. ^##^
*p* < 0.01 and ^#^
*p* < 0.05 compared with the control group. ** *p* < 0.01 and * *p* < 0.05 compared with the CTX group. ^§§^
*p* < 0.01, ^§^
*p* < 0.05, ^ns^
*p* > 0.05 compared with the GRT group.

**Figure 5 molecules-24-01096-f005:**
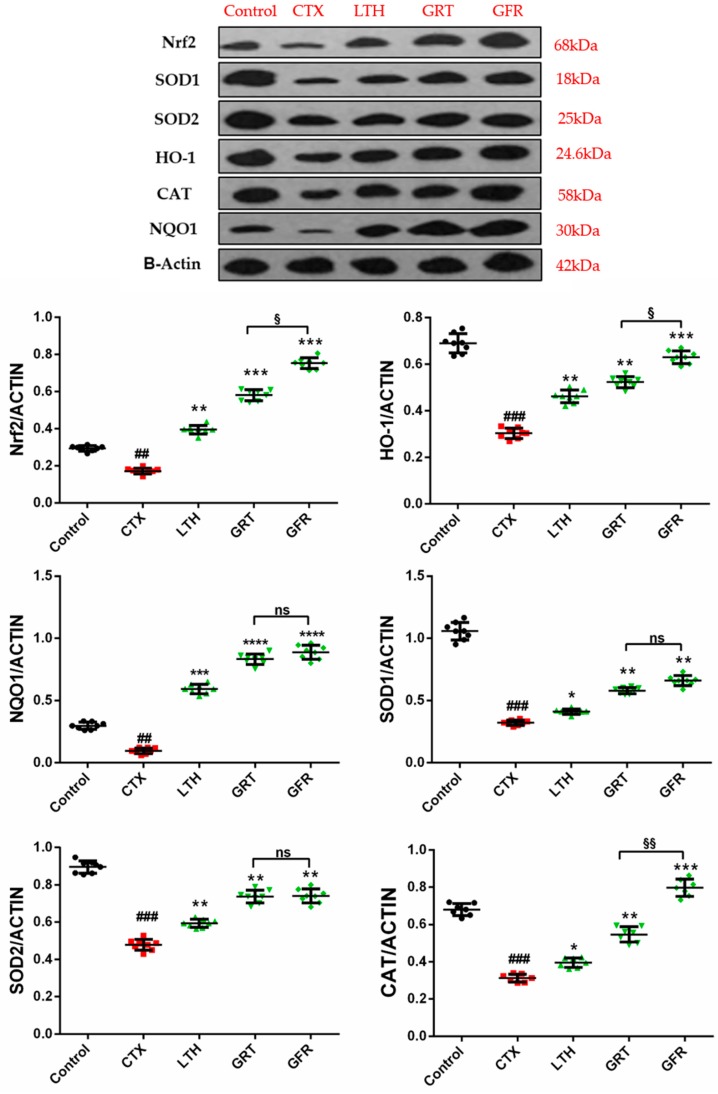
Effect of various parts of ginseng on the expression levels of nuclear factor-erythroid 2 related factor 2 (Nrf2), heme oxygenase-1 (HO-1), NADPH quinine oxidoreductase-1 (NQO1), superoxide-dimutase-1 (SOD1), superoxide-dimutase-2 (SOD2), and Catalase (CAT) in the spleen of mice with CTX-induced immunosuppression. The values are presented as mean ± SD, *n* = 10. ^###^
*p* < 0.001 and ^##^
*p* < 0.01 compared with the control group. *** *p* < 0.001, ** *p* < 0.01 and * *p* < 0.05 compared with the CTX group. ^§§^
*p* < 0.01, ^§^
*p* < 0.05, ^ns^
*p* > 0.05 compared with the GRT group.

**Table 1 molecules-24-01096-t001:** Effect of different parts of ginseng on body weight and immune organ indices in immunosuppressed mice.

Group	Initial Body Weight (g)	Final Body Weight (g)	Spleen Index (mg/g)	Thymus Index (mg/g)
	(*n* = 10)	(*n* = 10)	(*n* = 10)	(*n* = 10)
Control	19.84 ± 0.66	27.00 ± 1.41	4.16 ± 0.34	1.86 ± 0.12
CTX	22.07 ± 0.82	24.33 ± 1.28 ^#^	1.79 ± 0.21 ^##^	0.29 ± 0.05 ^##^
LTH	22.62 ± 1.22	26.29 ± 1.00 *	2.11 ± 0.30	0.33 ± 0.12
GRT	22.27 ± 1.38	24.92 ± 2.04	2.29 ± 0.26 **	0.49 ± 0.17 *
GSM	20.15 ± 1.51	23.12 ± 0.88	2.01 ± 0.18	0.29 ± 0.08
GLF	22.53 ± 2.23	24.62 ± 3.53	2.14 ± 0.21 *	0.39 ± 0.06 *
GFR	21.33 ± 1.76	23.58 ± 3.38	2.31 ± 0.23 * *	0.49 ± 0.23 *
GSD	23.98 ± 1.81	27.36 ± 2.70 *	2.03 ± 0.25	0.45 ± 0.12 *

Spleen index: spleen weight/body weight. Thymus index: thymus weight/body weight. The data are presented as the means ± SD. They were analyzed by one-way analysis of variance (ANOVA) test, followed by the least-significant difference post hoc test between multiple groups. ^#^
*p* < 0.05, ^##^
*p* < 0.01 compared with the control group; * *p* < 0.05, ** *p* < 0.01 compared with the CTX group. CTX, cyclophosphamide; LTH, levamisole hydrochloride tablet; GRT, ginseng root; GSM, ginseng stem; GLF, ginseng leaf; GFR, ginseng flower; GSD, ginseng seed.

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
