# Peer review of "A Comparative Study on the Effects of Different Parts of Panax ginseng on the Immune Activity of Cyclophosphamide-Induced Immunosuppressed Mice"

_molecules, 2019, doi:10.3390/molecules24061096_

Round 1
Reviewer 1 Report
In general this manuscript contains a lot of data on the action of ginseng but is underprepared
- Section 2.1 / table 1 : it is unclear whether the LTH, GRT groups were treated with CTX, the table etc should be more clear. Also how long animals were treated is not indicated. When spleen index and thymus index were determined (probably at the end of the experiment) is not indicated, thus this section needs more work
- dynamite plunger diagrams are better replaced by figures showing individual datapoints.
- Western blot in figure requires labeling and indication as to how representative it is
- Minor
Overall presentation needs rigorous checking, many small mistakes like line 94/95 - superfluous concanavalin A, or line 123 (Figure A2 [should be figure 2a], line 137) etc.
Discussion -
The immunostimulatory effect of Ginseng may be useful for treating diseases of which the pathology is dependent on reduced phagocytotic capacity like Crohn's disease ( Tschurtschenthaler M, Adolph TE, Ashcroft JW, Niederreiter L, Bharti R, Saveljeva S, Bhattacharyya J, Flak MB, Shih DQ, Fuhler GM, Parkes M, Kohno K, Iwawaki T, Janneke van der Woude C, Harding HP, Smith AM, Peppelenbosch MP, Targan SR, Ron D, Rosenstiel P, Blumberg RS, Kaser A. Defective ATG16L1-mediated removal of IRE1α drives Crohn's disease-like ileitis. J Exp Med. 2017 Feb;214(2):401-422; Parikh K, Zhou L, Somasundaram R, Fuhler GM, Deuring JJ, Blokzijl T, Regeling A, Kuipers EJ, Weersma RK, Nuij VJ, Alves M, Vogelaar L, Visser L, de Haar C, Krishnadath KK, van der Woude CJ, Dijkstra G, Faber KN, Peppelenbosch MP. Suppression of p21Rac signaling and increased innate immunity mediate remission in Crohn's disease. Sci Transl Med. 2014 Apr 23;6(233):233ra53)
Author Response
In general this manuscript contains a lot of data on the action of ginseng but is underprepared
Comment1: Section 2.1 / table 1 : it is unclear whether the LTH, GRT groups were treated with CTX, the table etc should be more clear. Also how long animals were treated is not indicated. When spleen index and thymus index were determined (probably at the end of the experiment) is not indicated, thus this section needs more work
Response: Thank you, dear reviewer. The model control (CTX), LTH, GRT, GSM, GLF, GFR and GSD groups were subjected to intraperitoneal injection of cyclophosphamide at a dosage of 50 mg/kg on Days 26–29, and mice were orally administration different parts of ginseng for 30 days. Spleens were collected immediately from the sacrificed mice under aseptic conditions.
Comment2: dynamite plunger diagrams are better replaced by figures showing individual datapoints.
Response: Thank you for your suggestion. All the figures have been revised into dynamite plunger diagrams for showing individual datapoints.
Comment3: Western blot in figure requires labeling and indication as to how representative it is.
Response: Thank you, dear reviewer. Western blot in figure and figure legend have added label and indication.
Comment4: Overall presentation needs rigorous checking, many small mistakes like line 94/95 - superfluous concanavalin A, or line 123 (Figure A2 [should be figure 2a], line 137) etc.
Response: Thank you for your careful check. Superfluous concanavalin A has been deleted, and the Figure 2 has been revised.
Comment5: The immunostimulatory effect of Ginseng may be useful for treating diseases of which the pathology is dependent on reduced phagocytotic capacity like Crohn's disease
( Tschurtschenthaler M, Adolph TE, Ashcroft JW, Niederreiter L, Bharti R, Saveljeva S, Bhattacharyya J, Flak MB, Shih DQ, Fuhler GM, Parkes M, Kohno K, Iwawaki T, Janneke van der Woude C, Harding HP, Smith AM, Peppelenbosch MP, Targan SR, Ron D, Rosenstiel P, Blumberg RS, Kaser A. Defective ATG16L1-mediated removal of IRE1α drives Crohn's disease-like ileitis. J Exp Med. 2017 Feb;214(2):401-422; Parikh K, Zhou L, Somasundaram R, Fuhler GM, Deuring JJ, Blokzijl T, Regeling A, Kuipers EJ, Weersma RK, Nuij VJ, Alves M, Vogelaar L, Visser L, de Haar C, Krishnadath KK, van der Woude CJ, Dijkstra G, Faber KN, Peppelenbosch MP. Suppression of p21Rac signaling and increased innate immunity mediate remission in Crohn's disease. Sci Transl Med. 2014 Apr 23;6(233):233ra53)
Response: Thank you, dear reviewer. Your suggestion is very meaningful and far-sighted for the deep study of immunosuppression effects of ginseng of various parts in CTX-induced, and we have showed the immunostimulatory effect of Ginseng may be useful for treating diseases of which the pathology is dependent on reduced phagocytotic capacity like Crohn's disease in the discussion (line240-242).

Reviewer 2 Report
The manuscript by Chen et al. examines the effects of different portions of the ginseng plant on immune responses using a variety of well-characterized in vivo and ex vivo techniques. Overall the paper is well-written and understandable. I have one major concern and several minor suggestions which should be addressed.
Major Concern:
(1) The authors compare up to 8 different groups using a one-way ANOVA. This necessitates 28 individual between-group comparisons, but the authors do not discuss any post-hoc adjustments to the p-value in their methods. The sheer number of comparisons greatly heightens the risk of type 1 error. Statistical control of type 1 error should be described in the methods, or data should be re-analyzed in the absence of previous post-hoc adjustments.
Minor Concerns:
(2) Line 62: Define Rd and Re
(3) Figures in the text are listed as A2, B2, etc. Generally accepted convention in most cases is 2A, 2B, etc.
(4) The conclusion section would be better placed after the discussion and before the methods section.
(5) It is unclear if different mice were used for each assay, or if mice were used for multiple assays.
(6) Line 404, report centrifuge in xg/RCF rather than RPM.
Author Response
Document 2. Responses to reviewer 2’s comments
Comments and Suggestions for Authors
The manuscript by Chen et al. examines the effects of different portions of the ginseng plant on immune responses using a variety of well-characterized in vivo and ex vivo techniques. Overall the paper is well-written and understandable. I have one major concern and several minor suggestions which should be addressed.
Major Concern:
Comment1: The authors compare up to 8 different groups using a one-way ANOVA. This necessitates 28 individual between-group comparisons, but the authors do not discuss any post-hoc adjustments to the p-value in their methods. The sheer number of comparisons greatly heightens the risk of type 1 error. Statistical control of type 1 error should be described in the methods, or data should be re-analyzed in the absence of previous post-hoc adjustments.
Response: Thank you, dear reviewer. We are very grateful to reviewer for the valuable comments and suggestions, which will help to improve the quality of the paper. In fact, we used one-way analysis of variance, and a post-hoc test was performed for intergroup comparisons using Bonferroni multiple comparison with the GraphPad Prism 6.0 software. We choose this method based on the following reason: Firstly, the number of samples in each group was the same. Secondly, because this experiment is hypothetical, we only need to compare each dose group with control group. Finally, compared with LSD, despite the low Bonferroni’s power is lower, but it can effectively avoid the 1 class error. Considering that the main purpose of the pharmacological experiments we studied was to reflect the significant differences in the treatment of ginseng in different parts, so we did not notice the need to discuss post-hoc adjustments in the paper. In addition, in order to avoid readers understanding the error, the data analysis method has been added in Statistical Analysis part. Thank you very much again for the reviewer’ comments, let us understand the deeper knowledge of statistics, which is very helpful for our future research.
Comment2: Line 62: Define Rd and Re
Response: Thank you for your suggestion. In this paper, the Rd and Re have been defined as ginsenoside.
Comment3: Figures in the text are listed as A2, B2, etc. Generally accepted convention in most cases is 2A, 2B, etc
Response: Thank you for your careful check. Figure 2 has been revised.
Comment4: The conclusion section would be better placed after the discussion and before the methods section.
Response: Thank you, dear reviewer. The conclusion section has been placed after the discussion and before the methods section.
Comment5: It is unclear if different mice were used for each assay, or if mice were used for multiple assays.
Response: Thank you, dear reviewer. The BALB/c mice from different batches of the same species were used in different assay.
Comment6: Line 404, report centrifuge in xg/RCF rather than RPM
Response: Thank you for your suggestion. The report centrifuge in xg/RCF has replaceed RPM (line427).
Round 2
Reviewer 2 Report
The authors have adequately addressed my previous comments
This manuscript is a resubmission of an earlier submission. The following is a list of the peer review reports and author responses from that submission.
Round 1
Reviewer 1 Report
The manuscript has been revised. However, the research progress on ginseng need to be summarized, such as: doi.org/10.1111/nyas.13424;doi: 10.1016/j.jgr.2015.12.004; doi: 10.1016/j.jpba.2014.12.005. In addition, the format of the reference list needs to be checked again.
Author Response
Comment: The manuscript has been revised. However, the research progress on ginseng need to be summarized, such as: doi.org/10.1111/nyas.13424;doi: 10.1016/j.jgr.2015.12.004; doi: 10.1016/j.jpba.2014.12.005. In addition, the format of the reference list needs to be checked again.
Response: Thank you for your suggestion. The research progress on ginseng has been perfected by reference doi.org/10.1111/nyas.13424;doi: 10.1016/j.jgr.2015.12.004; doi: 10.1016/j.jpba.2014.12.005 (line 38-42).
Reviewer 2 Report
In this manuscript, author to compare the effects of immunological activity from ginseng of various parts on immune system of immunosuppressive mice. It is interesting. However, some issues need to improve in this manuscript.
The quality of all figures needs to improve. In all of figure x-axis show ginseng of various parts group (GRT, GSM, GLF, GFR, and GSD). Whether all groups inject CTX not mention in the experiment section. Figure 5 WB figure need to include the molecular weight of protein and lack beta-actin band in this figure.
The dosage of ginseng of various parts group does not mention in this manuscript.
The preparation of ginseng extracts not include the reference in 4.2 section.
What kind of active chemical compound include in GRT and GFR need to the discussion in this manuscript.
Effect of IL-4 needs to include in the discussion section.
Page number need to include in this manuscript.
Author Response
In this manuscript, author to compare the effects of immunological activity from ginseng of various parts on immune system of immunosuppressive mice. It is interesting. However, some issues need to improve in this manuscript.
Comment1: The quality of all figures needs to improve. In all of figure x-axis show ginseng of various parts group (GRT, GSM, GLF, GFR, and GSD). Whether all groups inject CTX not mention in the experiment section. Figure 5 WB figure need to include the molecular weight of protein and lack beta-actin band in this figure.
Response: Thank you for your suggestion. The quality of all figures of the manuscript has attained 300 dpi, and all figure x-axis of ginseng of various parts group has showed GRT, GSM, GLF, GFR, and GSD,and CTX-challenge in all groups have been mentioned in the experiment section (line 313), and Figure 5 WB figure has included the molecular weight of protein and beta-actin band.
Comment2: The dosage of ginseng of various parts group does not mention in this manuscript.
Response: Thank you for your careful check. The dosage of ginseng of various parts group has been added in this manuscript (line 318).
Comment3: The preparation of ginseng extracts not include the reference in 4.2 section.
Response: Thank you for your careful check. The preparation of ginseng extracts have added the reference in 4.2 section (line 296).
Comment4: What kind of active chemical compound include in GRT and GFR need to the discussion in this manuscript.
Response: Thank you for your suggestion. The GRT and GFR include active chemical compound and the discussion have been showed in the manuscript (line 238-244) and supplementary materials (line 242-249).
Comment5: Effect of IL-4 needs to include in the discussion section.
Response: Thank you, dear reviewer. Your suggestion is very meaningful and far-sighted for the deep study of immunosuppression effects of ginseng of various parts in CTX-induced. The manuscript has added the effect of IL-4 in the discussion section (line 228-232 and 239-240).
Comment6: Page number need to include in this manuscript.
Response: Thank you for your careful check. The manuscript has added page number.

Reviewer 3 Report
Authors made cyclophosphamide-induced immunosuppression mouse model and assessed the effects of Ginseng root, ginseng leaf, ginseng flower. However, there is a critical concern to be published in ‘Molecule’.
In this study, the extracts of ginseng parts were used and evaluated. It is generally accepted that the effects of extracts are ambiguous and the reproducibility is low. Each extract (ginseng root, ginseng leaf, ginseng flower) may include different components with various concentrations Authors should select and focus more than one component, and try to explain the results
Author Response
Authors made cyclophosphamide-induced immunosuppression mouse model and assessed the effects of Ginseng root, ginseng leaf, ginseng flower. However, there is a critical concern to be published in ‘Molecule’.
Comment: In this study, the extracts of ginseng parts were used and evaluated. It is generally accepted that the effects of extracts are ambiguous and the reproducibility is low. Each extract (ginseng root, ginseng leaf, ginseng flower) may include different components with various concentrations Authors should select and focus more than one component, and try to explain the results
Response: Thank you, dear reviewer, and your suggestion is very meaningful. The immune effects of GRT, GLF and GFR groups were better than the GSM and GSD groups, which may due to the content of ginsenosides and amino acids of GRT, GLF and GFR were significantly higher than the GSM and GSD groups. And individual ginsenoside the content of Rg1, Re, Rb1, Rc and Rb2 in GRT, GFR and GSD groups were remarkably higher than the GSM and GSD groups. And the research demonstrated Rg1, Re, and Rb1 have more potent adjuvant properties, indicating that they are the major constituents contributing to the adjuvant activities of total ginseng saponins, and the response has been added the discussion of the manuscript (line 242-249).

Reviewer 4 Report
The manuscript is interesting.
1. Was the extract toxic? This is important to establish the safety of the product
2.The extract should contain a type of standardization (e.g. fingerprint)
3. In the Discussion, the Authors should highlight the possible clinical significance of their findings
Author Response
The manuscript is interesting.
Comment1: Was the extract toxic? This is important to establish the safety of the product
Response: Thank you, dear reviewer. The ginseng extract was not toxic, which has been recorded by books’ of “Ginseng, the Genus Panax” and “Textbook of Complementary and Alternative Medicine”.
Comment2: The extract should contain a type of standardization (e.g. fingerprint)
Response: Thank you for your suggestion. The supplementary materials have added the chromatograms of gensenosides and amino acid mixture standard (A) and ginseng sample (B), and the content of saponins, total polysaccharides, ginsenosides and amino acids of the GRT, GSM, GLF, GFR, GSD, which the results have been accepted by Food Science.
Comment3: In the Discussion, the Authors should highlight the possible clinical significance of their findings
Response: Thank you for your suggestion. The manuscript have summarized the development and application of the findings in conclusion section (line 427-430).

Round 2
Reviewer 2 Report
Can be accepted publish in "Molecules" now.
Reviewer 3 Report
In Author's response, I can't find the exact answer for my last comments. The effects of plant extract are very ambiguous with low reproducibility. Authors should focus on more than one active component in each extract and measure their amounts/biological effects.